

# Shallow Cumulus Cloud Feedback in Large Eddy Simulations - Bridging the Gap to Storm Resolving Models

Jule Radtke[1,2], Thorsten Mauritsen[3], and Cathy Hohenegger[4]

[1]Meteorological Institute, Center for Earth System Research and Sustainability (CEN), Universität Hamburg, Hamburg, Germany
[2]International Max Planck Research School on Earth System Modelling, Max Planck Institute for Meteorology, Hamburg, Germany
[3]Department of Meteorology, Stockholm University, Stockholm, Sweden
[4]Max Planck Institute for Meteorology, Hamburg, Germany

**Correspondence:** Jule Radtke (jule.radtke@uni-hamburg.de)

**Abstract.** The response of shallow trade cumulus clouds to global warming is a leading source of uncertainty to interpretations and projections of the Earth's changing climate. A setup based on the Rain In Cumulus over the Ocean field campaign is used to simulate a shallow trade wind cumulus field with the Icosahedral Non-hydrostatic Large Eddy Model in a control and a perturbed 4K warmed climate, while degrading horizontal resolution from 100 m to 5 km. As the resolution is coarsened the

basic state cloud fraction increases substantially, especially at cloud base, lateral mixing is weaker and cloud tops reach higher. Nevertheless, the overall vertical structure of the cloud layer is surprisingly robust across resolutions. In a warmer climate, cloud cover reduces, alone constituting a positive shortwave cloud feedback: the strength correlates with the amount of basic state cloud fraction, thus is stronger at coarser resolutions. Cloud thickening, resulting from more water vapor availability for condensation in a warmer climate, acts as a compensating feedback, but unlike the cloud cover reduction it is largely

resolution independent. Therefore, refining the resolution leads to convergence to a near-zero shallow cumulus feedback. This dependence holds in experiments with enhanced realism including precipitation processes or warming along a moist adiabat instead of uniform warming. Insofar as these findings carry over to other models, they suggest that storm resolving models may exaggerate the trade wind cumulus cloud feedback.

## 1 Introduction

How shallow cumulus clouds respond to global warming has been recognized as a critical source of uncertainty to process- or model-based estimates and interpretations of the Earth's changing climate (Bony and Dufresne, 2005; Vial et al., 2013; Zelinka et al., 2020; Flynn and Mauritsen, 2020). Most frequently shallow cumulus clouds are observed in the trade wind region and thus often called trade-wind cumuli, even if they appear in most regions on Earth. Due to their widespread occurence over the world's oceans, shallow cumuli are, though small in size, crucial to the Earth's radiative balance and act to cool the

Earth by reflecting shortwave radiation (Hartmann et al., 1992). Their response to global warming is therefore important for the global-mean cloud feedback. Actually, it is their differing response to warming that explains much of the disagreement



in climate sensitivity across climate models (Bony and Dufresne, 2005; Webb et al., 2006; Vial et al., 2013; Boucher et al., 2013; Medeiros et al., 2015; Zelinka et al., 2020; Flynn and Mauritsen, 2020). Most global climate models (GCMs) simulate a positive low cloud feedback primarily due to reduction of cloud cover in response to warming. In models probed in the fifth

phase of the Coupled Model Intercomparison Project (CMIP5) the low-level cloud feedback varies between 0.16 to 0.94 $\mathrm{Wm}^{-2}$ with most spread coming from the low-cloud amount feedback, the latter with values ranging between -0.09 and 0.63 $\mathrm{Wm}^{-2}$ (Boucher et al., 2013; Zelinka et al., 2016).

Emerging tools to advance understanding are global high resolution models that unlike climate models explicitly simulate
convective motions instead of parameterizing them (Stevens et al., 2020). In past studies of shallow cumulus clouds and their response to a warmer climate mostly large eddy (hectometer resolving) simulations (LES) have been used (Rieck et al., 2012; Blossey et al., 2013; Bretherton et al., 2013; Vogel et al., 2016; Stevens et al., 2001; Siebesma et al., 2003; van Zanten et al., 2011). LES is a turbulence modeling technique in which most of the energy containing motions are explicitly resolved, but because of their computational expense LES studies have been limited in their domain size and timescales. Due to increasing
computational power, it has become possible to run simulations on global domains, albeit not at hectometer but kilometer scales (e.g. Tomita, 2005; Stevens et al., 2019). These models are often called cloud resolving or convection permitting models (Prein et al., 2015) but here referred to as storm resolving models (SRMs) following Klocke et al. (2017) and Stevens et al. (2019); see also Satoh et al. (2019) for a discussion of naming. Global SRMs provide the opportunity to study cloud feedbacks without having to rely on an uncertain convective parameterization and while interacting with the large scale environment, but
at a typical grid spacing of a few kilometers shallow convection is allegedly poorly resolved.

This study aims to bridge the gap between findings based on limited-area large eddy simulations that typically use hectometer or finer grid spacings and emerging global storm resolving models that apply kilometer grid spacings. It investigates how the representation of shallow cumuli and their climate feedback is affected by the choice of horizontal resolution. To do so a setup
based on the Rain In Cumulus over the Ocean field campaign is used (Rauber et al., 2007). A shallow trade cumulus field is simulated with the Icosahedral Non-hydrostatic Large Eddy Model (Dipankar et al., 2015; Heinze et al., 2017) in a control and a perturbed 4K warmed climate while degrading horizontal resolution from 100 m to 5 km. The results are discussed by initially looking at the effect of resolution on the representation of shallow cumulus clouds in a control climate in Sect. 3, subsequently on the response of shallow cumulus clouds to a warming climate in Sect. 4.

## 50    2    Model and Setup

Experiments are conducted with the ICOsahedral Non-hydrostatic Large Eddy Model (ICON-LEM). ICON was developed in collaboration between the Max Planck Institute for Meteorology and the German Weather Service and solves the equations of motions on an unstructured triangular Arakawa C grid. For global applications, it is based on successive refinement of a spherical icosahedron (Zängl et al. 2015), but here a two-way cyclic torus domain is used. A detailed description of the LES version





ICON-LEM can be found in Dipankar et al. (2015). In the specific ICON-LEM setup for this study subgrid scale turbulence
is modeled based on the classical Smagorinsky scheme with modifications by Lilly (1962). For microphysical properties, the
simple saturation adjustment scheme is used in experiments where precipitation is prohibited. In experiments with precipitation
processes the two-moment mixed-phase microphysics scheme of Seifert & Beheng (2006) is applied. Radiation is computed
with the Rapid Radiation Transfer Model scheme (RRTM, Mlawer et al. 1997). A simple all-or-nothing scheme is applied for
cloud fraction (Sommeria & Deardorff 1977).

The setup is based on the Rain In Cumulus over the Ocean (RICO) measurement campaign (Rauber et al., 2007). The RICO
case developed by van Zanten et al. (2011) prescribes large scale forcings and initial profiles characteristic of the broader trades
and serves as a control experiment representative of present climate conditions. Figure 1 shows the profiles used for initializa-
tion of potential temperature $\theta$, specific humidity $q_v$ and the horizontal winds $u$ and $v$. The large scale forcing is prescribed
with time-invariant profiles of the subsidence rate and temperature and moisture tendencies due to radiative cooling and hor-
izontal advection. As modification to the case defined by van Zanten et al. (2011), radiation is computed interactively to be
able to calculate cloud radiative effects, which requires a model top of about 20 km in ICON-LEM. Below 4 km height, initial
profiles and large scale forcings as in van Zanten et al. (2011) are applied, above they are expanded accordingly, mostly with
piecewise linear extrapolation, see Appendix A1 for details. Sea surface temperature is fixed at 299.8 K as in the RICO set-up,
and bulk aerodynamics formulas parameterize the surface momentum and thermodynamic fluxes. Simulations are performed
on a pseudo-Torus grid with doubly periodic boundary conditions and flat geometry. The domain is fixed over a central latitude
of 18°N. In the vertical 175 levels are used with grid spacings of 40 to 60 m beneath 5 km height stretching to approximately
300 m at the model top of 22 km. Duration of the simulations is 48 hours and statistics shown are the second day mean.

75

The warming experiment design follows a simple idealistic climate change as used in e.g. Rieck et al. (2012). It increases
the temperature profile compared to the control run while keeping relative humidity constant. Simulations are run with five
different horizontal resolutions, 100 m, 500 m, 1 km, 2.5 km and 5 km, employed on three different domain sizes. The domain
sizes are chosen to be ideally suitable to run with two different horizontal resolutions. They span 50 to 200 points resulting
80 in domain sizes between 12 x 12 km and 500 x 500 km. The basic experiment inhibits precipitation and warms surface and
atmosphere uniformly by 4 K as in Rieck et al. (2012). Furthermore two refined experiments are conducted, one allowing
precipitation to develop (e.g. as in Vogel et al., 2016), and another one altering the vertical warming to follow a moist adiabat
(e.g. as in Blossey et al., 2013). These tests show how robust the findings are against simplifications made in the original
experimental setup. See Table 1 for an overview of the different experiments.



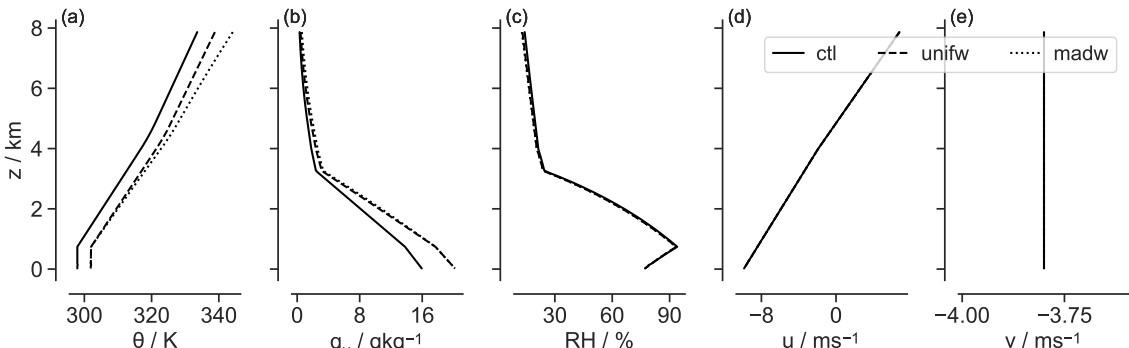

**Figure 1.** Initial profiles of (a) potential temperature $\theta$, (b) specific humidity $q_v$, (c) relative humidity $RH$, (d-e) horizontal winds $u$ and $v$ for the control (solid line) and the perturbed (vertically uniform warming, dashed line and warming following a moist adiabat, dotted line) climate states.

**Table 1.** Specifications used for the different pertubation experiments. Specific humidity in the perturbed runs (unifw and madw) is adjusted to keep the relative humidity constant compared to the control simulation.

| Hor. Resolution | Hor. Domain | Gridpoints | Temp profile | Prec | Casename |
|---|---|---|---|---|---|
| 100m | 12.6 x 12.6 km$^2$ | 126$^2$ | control | no, yes | 100m.ctl, -P |
| | | | +4 K | no, yes | 100m.unifw, -P |
| | | | +4 K moist adiabatic | no | 100m.madw |
| 500m | 50 x 50 km$^2$ | 100$^2$ | control | no, yes | 500m.ctl, -P |
| | | | +4 K | no, yes | 500m.unifw, -P |
| | | | +4 K moist adiabatic | no | 500m.madw |
| 1km | 50 x 50 km$^2$ | 50$^2$ | control | no | 1km.ctl |
| | | | +4 K | | 1km.unifw |
| 2.5km | 500 x 500 km$^2$ | 200$^2$ | control | no | 2.5km.ctl |
| | | | +4 K | | 2.5km.unifw |
| 5km | 500 x 500 km$^2$ | 100$^2$ | control | no, yes | 5km.ctl, -P |
| | | | +4 K | no, yes | 5km.unifw, -P |
| | | | +4 K moist adiabatic | no | 5km.madw |
| *Additional sensitivity experiment:* | | | | | |
| 1km | 500 x 500 km$^2$ | 500$^2$ | +4 K | no | large |





## 3 Basic state dependency on resolution

In this Section we present characteristics of the simulated shallow cumulus regime in the control case and highlight similarities and differences as the resolution is coarsened. This lays out the ground to study in the following how shallow cumulus clouds respond to a perturbed warmer climate and how this depends on horizontal resolution in Sect. 4.

### 3.1 Standard Case

At 100 m resolution a typical trade wind cumuli field is simulated that is in line with the range of LES analyzed in the RICO LES intercomparison case (van Zanten et al., 2011). Total cloud cover is 15 % (Fig. 2) which is slightly lower than the cloud cover of 17 % observed during the RICO field study (Nuijens et al., 2009) and the ensemble mean cloud cover of 19 % in the RICO intercomparison case (range 9 - 38 %). The vertical structure is consistent with the general picture of trade wind cumuli cloud layers (Fig. 3). Cloud fraction peaks at cloud base (6 %) near 700 m , then decreases sharply with height, thereafter keeping a value of about 2% through the cumulus layer until 2 km (Fig. 3). Above this height, cloud fraction increases again due to detrainment at cloud top before declining sharply under the trade inversion at around 2.5 km height. Temperature increase and sharp humidity decrease mark the inversion and top of the cloud layer.

At coarser resolutions the overall structure of the boundary and cloud layer is surprisingly similar to the 100 m resolution simulation. The vertical structure of cloud fraction is in all experiments characterized by a dominant peak at cloud base and a second smaller peak near the inversion (Fig. 3). Therefore, at all resolutions cloudiness at cloud base contributes most to total cloud cover. All experiments simulate a well-mixed subcloud layer, a transition layer which is most evident in the moisture gradients, a cloud layer, and an inversion layer into which the clouds penetrate and detrain (Fig. 4). However, at coarser resolutions the transition layer is more pronounced exhibiting a stronger moisture gradient and the inversion height is more distributed in the vertical. These variations translate into the most notable differences between the resolutions.

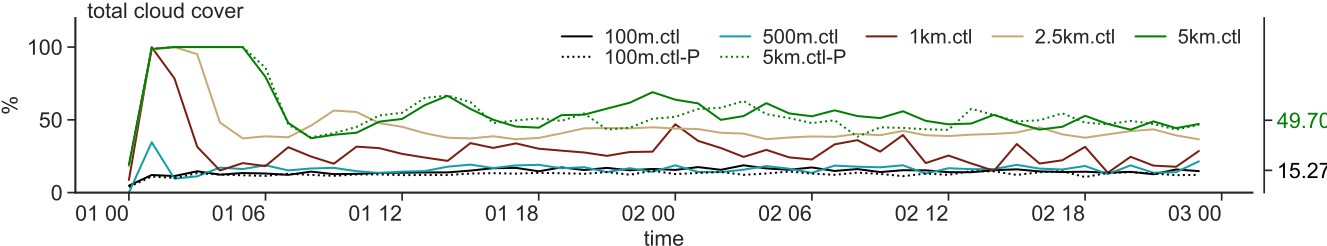

**Figure 2.** Temporal evolution of total cloud cover in ctl at 100 m, 500 m, 1 km, 2.5 km and 5 km resolution (solid lines) and ctl-P at 100 m and 5 km resolution (dotted lines). Ordinates on the right axis display the second day domain averaged total cloud cover for 100m.ctl and 5km.ctl (see Table 2 and 3 for more statistics).

Most importantly, we note that at coarser resolutions cloud cover is substantially enhanced (Fig. 2). At 5 km resolution total cloud cover is more than three times higher than at 100 m (50 vs 15 %). This increase in cloud cover is mostly due to enhanced





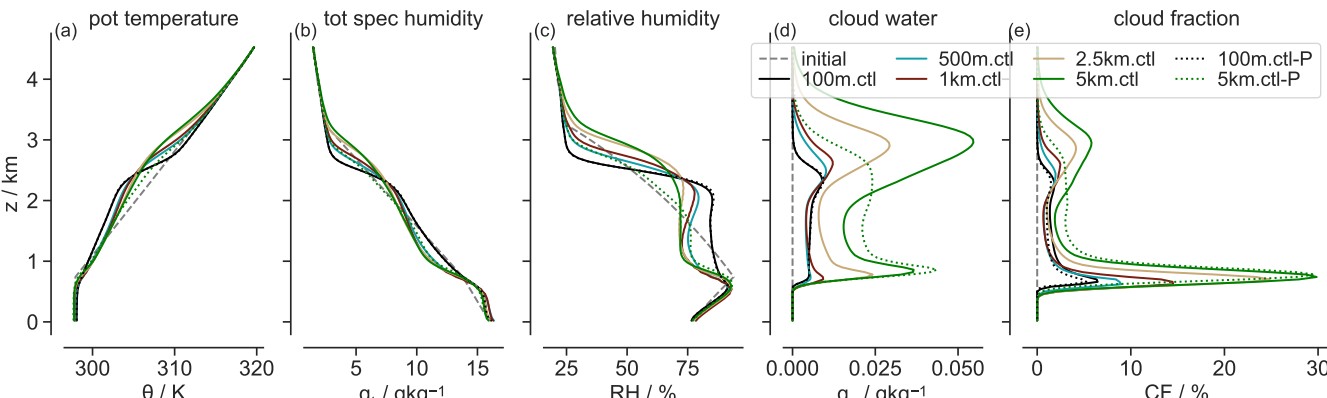

**Figure 3.** Profiles of second day domain averaged (a) potential temperature $\theta$, (b) total water specific humidity $q_t$, (c) relative humidity $RH$, (d) cloud water $q_c$ and (e) cloud fraction $CF$ for different horizontal resolutions of the ctl (solid lines) and ctl-P simulations (dotted lines).

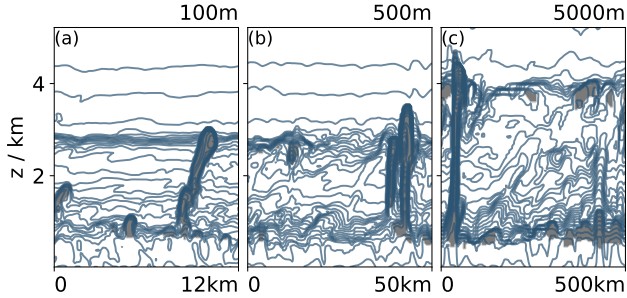

**Figure 4.** Cross section (note the different horizontal extent) of total water specific humidity field and cloud cover at 42 hours simulation time in the ctl simulations for three different horizontal resolutions: (a) 100 m, (b) 500 m, (c) 5 km. The total water specific humidity field is shown as contours evenly spaced every $0.5\,\mathrm{g\,kg^{-1}}$ and cloud fraction is grey shaded.

cloudiness at cloud base and to a smaller extent from an increase in cloud fraction near the inversion (Fig. 3). The ratio between cloudiness at cloud base and total cloud cover rises from 0.4 with the 100 m to 0.6 with the 5 km resolution, that is, cloud base cloud fraction contributes more to total cloud cover in the coarser resolution simulations. Further, at coarser resolutions clouds reach higher (Fig. 3). At 5 km resolution clouds deepen up to an inversion height of about 3.2 km, around 700 m higher than at the finest resolution. Both characteristics can be confidently linked to resolution and not domain size as a sensitivity experiment shows (see Appendix B1).

Larger cloud cover and higher cloud tops at coarser resolutions can be attributed to weaker small-scale mixing. At coarse resolutions the subcloud layer ventilates less efficiently and the subcloud and cloud base layer are therefore moister and cooler and as a result associated with stronger surface sensible but weaker latent heat fluxes (Table 2). Moister and colder conditions are consistent with weaker cumulus massfluxes and weaker entrainment of warm dry air from aloft. Because conditions are moister





**Table 2.** Averages of total cloud cover ($CC$), maximum vertical cloud fraction ($CF_{max}$), liquid water path ($LWP$), surface sensible heat flux ($SH$), surface latent heat flux ($LH$), inversion height ($z_i$, representening the location of maximum $\theta$-gradients), cloud base height ($z_b$, representing the minimum height where 50 % of $CF_{max}$ is reached) and change in the shortwave cloud radiative effect $\Delta SWCRE$ at 100 m, 500 m and 5 km resolutions in the non-precipitating simulations of the ctl, unifw and madw climate states.

| | Case | $CC$ | $CF_{max}$ | $LWP$ | $SH$ | $LH$ | $z_i$ | $z_b$ | $\Delta SWCRE$ |
|---|---|---|---|---|---|---|---|---|---|
| | | % | % | gm$^{-2}$ | Wm$^{-2}$ | Wm$^{-2}$ | m | m | Wm$^{-2}$ |
| | ctl | 15.27 | 6.46 | 12.06 | 4.49 | 153.57 | 2560 | 610 | |
| 100m | unifw | 14.28 | 6.11 | 13.65 | 3.40 | 199.86 | 2810 | 670 | 0.21 |
| | madw | 14.24 | 6.21 | 13.42 | 3.31 | 190.61 | 2660 | 640 | 0.19 |
| | ctl | 16.60 | 8.94 | 13.70 | 5.79 | 140.81 | 2760 | 570 | |
| 500m | unifw | 13.22 | 6.90 | 15.77 | 4.85 | 182.75 | 3090 | 600 | 0.47 |
| | madw | 13.56 | 7.08 | 16.02 | 4.58 | 174.74 | 2810 | 580 | 0.32 |
| | ctl | 51.21 | 29.85 | 86.54 | 7.26 | 149.09 | 3240 | 610 | |
| 5km | unifw | 42.36 | 23.69 | 90.52 | 6.33 | 180.51 | 3570 | 640 | 6.3 |
| | madw | 43.30 | 24.38 | 84.62 | 5.70 | 177.61 | 3180 | 620 | 6.6 |

and colder in the boundary layer, relative humidity is enhanced and saturation is more likely leading to more widespread cloud formation at coarser resolutions. Hohenegger et al. (2020) found similar characteristics in global simulations with explicit convection and grid spacings ranging between 2.5 and 80 km.

Additionally, at coarser resolutions small-scale lateral mixing between cumulus clouds and their environment is markedly weaker which explains the higher cloud tops. Figure 5 displays the fractional entrainment and detrainment rates as a measure for lateral mixing intensity diagnosed after Stevens et al. (2001). The entrainment rate at 100 m resolution decreases from $2\,\mathrm{km}^{-1}$ near cloud base to $1.2\,\mathrm{km}^{-1}$ in the cloud layer, which is similar to the rates found in the RICO LES intercomparison case (van Zanten et al., 2011). At 500 m resolution the mean entrainment rate in the cloud layer is around $0.8\,\mathrm{km}^{-1}$, in 5 km around $0.4\,\mathrm{km}^{-1}$, thus notably weaker. This might be attributed to larger cloud structures that offer less surface area for dilution compared to smaller cloud structures that are resolved at finer resolutions. Because they dilute less, clouds retain more buoyancy and reach higher at coarser resolutions.

### 3.2 Precipitating Case

Trade wind cumulus clouds rain frequently as observations show (Nuijens et al., 2009). We activate precipitation processes to test if the identified resolution dependence is robust in simulations with 100 m, 500 m and 1 km horizontal resolution.

We find that including precipitation processes mainly acts to limit cloud layer deepening. Whereas the 100 m resolution simulations are very similar, with 500 m resolution the inversion height in the precipitating case is around 150 m and with





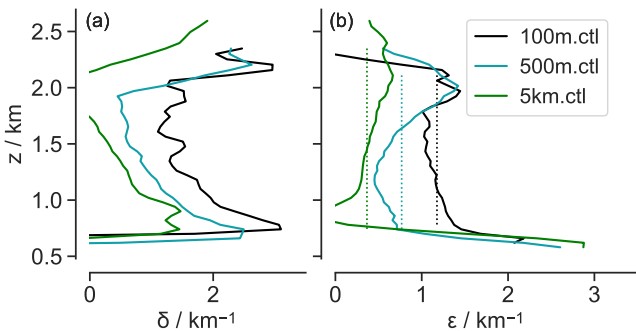

**Figure 5.** (a) Fractional entrainment ($\varepsilon$) and (b) detrainment rate ($\delta$) at 100 m, 500 m and 5 km resolution in CTL. The mean entrainment rate in the cloud layer is shown as dotted lines.

5 km around 350 m lower than in the non-precipitating case (Table 3). In the RICO LES intercomparison case van Zanten et al. (2011) also found that precipitating simulations with 100 m resolution cause an approximate 100 m reduction in the depth of the cloud layer. Precipitation acts to limit cloud layer deepening because it removes moisture available for evaporation near the inversion. The precipitating cloud field is therefore also characterized by less cloud fraction near the inversion (Fig. 3).

**Table 3.** As in Tab. 2 but for the precipitating simulations (P) of the CTL and +4K climate states.

| | Case | $CC$ | $CF_{max}$ | $LWP$ | $SH$ | $LH$ | $z_i$ | $z_b$ | $\Delta SWCRE$ |
|---|---|---|---|---|---|---|---|---|---|
| | | % | % | $gm^{-2}$ | $Wm^{-2}$ | $Wm^{-2}$ | m | m | $Wm^{-2}$ |
| 100m | ctl-P | 13.09 | 5.26 | 12.88 | 4.54 | 154.02 | 2580 | 610 | |
| | unifw-P | 12.38 | 4.72 | 13.62 | 3.42 | 200.05 | 2810 | 670 | 0.025 |
| 500m | ctl-P | 13.58 | 6.64 | 11.93 | 6.70 | 139.90 | 2630 | 580 | |
| | madw-P | 10.95 | 5.37 | 12.98 | 6.57 | 182.14 | 2780 | 610 | 0.16 |
| 5km | ctl-P | 49.63 | 29.59 | 55.73 | 7.95 | 147.10 | 2860 | 660 | |
| | madw-P | 44.91 | 25.61 | 51.57 | 7.69 | 180.05 | 2980 | 690 | 5.2 |


Furthermore, we find that the precipitating cloud fields exhibit more cloud fraction in the lower parts of the cloud layer as compared to the non-precipitating cloud field (Fig. 3). Vogel et al. (2016) and van Zanten et al. (2011) found a similar increase in cloud fraction and explained it by increased evaporation from precipitation concentrated in the cloud layer, noting that the

evaporation of precipitation must not be confined to the subcloud layer. Due to this moistening, latent heat fluxes are moderately weaker, e.g. at 5 km around 2 $Wm^{-2}$ (compare Tab. 2 and Tab. 3). Additionally, evaporation of falling raindrops induces a cooling in the subcloud layer, which results in stronger surface sensible heat fluxes.

Because liquid is removed through precipitation, and clouds are shallower, the precipitating simulations have a lower total

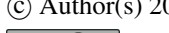



cloud cover than the non-precipitating simulations at all resolutions (15 vs 13% at 100 m, 16.6 vs 13.6% at 500 m, 51.2 vs
49.7% at 5 km; Tables 2 and 3). However, changes between the non-precipitating to precipitating cloud field are small and
additionally similar across resolution. Therefore, the resolution dependencies remain dominant in the precipitating case as in
the non-precipitating case: cloud cover is substantially enhanced and clouds are deeper at coarser resolutions.

## 4   Cloud response to warming across resolutions

Here, we investigate how the cloud field responds to warming in dependence of resolution. First, the response to a uniform
temperature shift in the standard non-precipitating case is discussed and how the resolution dependence of the basic state cloud
field affects the cloud field's response to warming. Second, the robustness of our results are investigated by testing whether
warming along a moist adiabat or in the precipitating case alters the response across resolution.

### 4.1   Response to uniform warming

At 100 m resolution we find a slight cloud cover reduction as response to uniform warming in line with earlier LES-based
studies (Rieck et al., 2012; Blossey et al., 2013; Vogel et al., 2016). Total cloud cover decreases from 15.3% to 14.3% (Table
2). It seems plausible that drying (Fig. 6), that results from mixing due to the stronger vertical gradient in specific humidity
within the warmer case, could explain much of this reduction in cloud cover (Bretherton, 2015; Brient and Bony, 2013). It has
further been suggested that enhanced surface latent heat fluxes envigorate convection, deepening the cloud layer and leading
to further drying by mixing (Stevens, 2007; Rieck et al., 2012). However, as more refined experiments (Sect. 4.2) do not result
in substantial deepening, this process appears to be of secondary importance. The cloud cover reduction on its own constitutes
a positive shortwave cloud feedback.

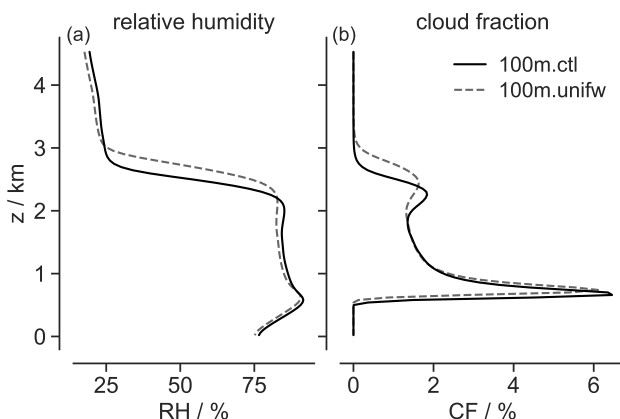

**Figure 6.** Profiles of second day domain averaged (a) relative humidity $RH$ and (b) cloud fraction $CF$ for the control (solid line) and
vertically uniform warmed (dashed line) simulation at 100 m resolution.





Also at coarser resolutions, we find cloud cover reductions as response to uniform warming (Table 2). Across resolutions
the cloud layer is drier, cloud cover reduced and cloud tops reach higher (Fig. 7). The magnitude of cloud cover reduction,
however, differs: at 100 m resolution total cloud cover reduces by 1% point, whereas at 5 km resolution total cloud cover
reduces by roughly 9% points. At coarse resolutions it is distinctly cloud base cloudiness that reduces with warming. This
low resolution behavior is in contrast to the results of previous high resolution LES studies and observations which suggest a
relatively invariant cloud base fraction (Nuijens et al., 2014; Siebesma et al., 2003), but is a common feature in global climate
model simulations (Brient and Bony, 2013; Brient et al., 2015; Vial et al., 2016). We find that the strength of cloud reduction
correlates well with the amount of cloud cover in the basic state (Fig. 8). The more clouds are present in the basic state, the
more cloudiness reduces in the warmer climate. Hence, because cloud cover increases at coarser resolutions, in particular near
cloud base, they show a stronger cloud reduction than at high resolutions.

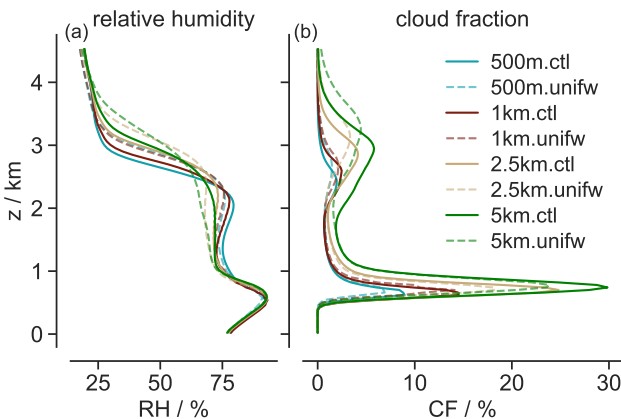

**Figure 7.** As in Fig. 6 but for 500 m, 1 km, 2.5 km and 5 km resolution.

From the reduction in cloud amount, a positive shortwave feedback would be expected, however, the total shortwave feedback
at high resolutions is close to zero, e.g. at 100 m with a value of $0.05\,\mathrm{Wm^{-2}K^{-1}}$ (Fig. 9). This is due to a compensating
feedback from cloud thickening. The cloud liquid water path increases at all resolutions with warming (Fig. 9). Clouds become
more reflective contributing a negative shortwave feedback. In contrast to the cloud amount reduction though, cloud thickening
is not strongly resolution dependent. An increasing cloud water content with warming is to be expected as more water vapor is
available for condensation (Paltridge, 1980); an argument that is not reliant in any meaningful way on resolution. Consequently,
the total shortwave feedback shows the same dependence on resolution as the cloud reduction and correlates well with the basic
cloud cover, too (Fig. 8). Hence the shortwave cloud feedback is weak or close to zero for high resolution and positive for coarse
resolutions.





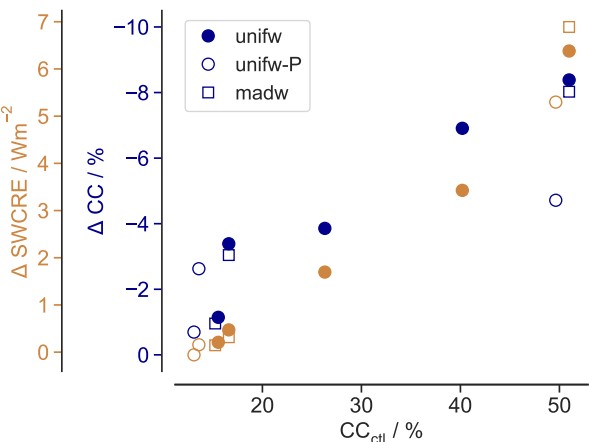

**Figure 8.** Relationship between cloud cover amount in the control simulation ($CC_{ctl}$) and cloud cover reduction with warming ($\Delta CC$) as well as the shortwave cloud radiative feedback ($\Delta SWCRE$) across all simulations.

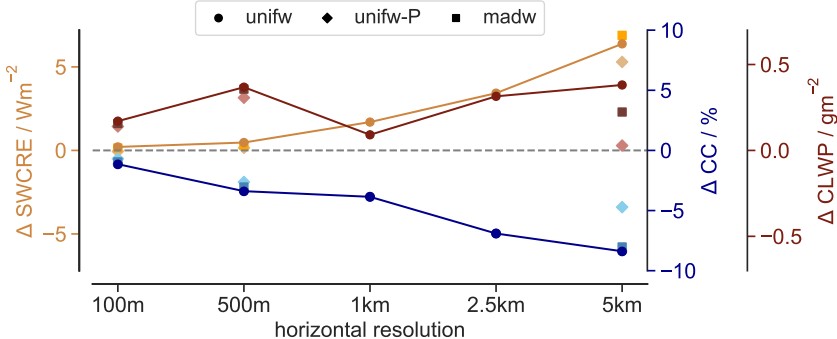

**Figure 9.** Shallow cumulus cloud feedback across resolution: Shortwave cloud radiative feedback $\Delta SWCRE$, change in cloud cover amount $\Delta CC$ and cloud liquid water path $\Delta CLWP$ between the pertubed warmer and control simulations for all experiments.

## 4.2 Sensitivity of response to refined experimental setups

The base case studied above was admittedly simplistic in that there is no precipitation and a vertically uniform warming was applied. Here we explore the effects of these assumptions. The free tropospheric temperature profile in the Tropics is set by the regions of deep convection that are close to a moist adiabat. Therefore, the tropical temperature is expected to warm close to a moist adiabat, leading to more warming aloft than at the surface and has been used in other modelling studies (e.g Blossey et al., 2013; Bretherton et al., 2013). With moist adiabatic warming an increase in dry static stability is introduced: the initial

lower tropospheric stability ($LTS = \theta_{700} - \theta_0$) increases from 13.1 K to 14.4 K, and as a result, with moist adiabatic warming the cloud response near the trade inversion is muted (Fig. 10). Both the cloud top height and cloud fraction in the upper re-





gions change only little. The inversion height in the moist adiabatic warming case varies compared to the control case by only around 50 to 100 m, whereas in the uniform warming case the inversion height increased markedly by around 300 m (Table 2). Therefore, cloud deepening is at all resolutions slightly weaker. Nevertheless, total cloud cover reduction is only slightly 200 dampened. (Fig. 9). Overall the changes are small, though, and therefore, the total shortwave cloud radiative feedbacks is only slightly reduced when applying the more realistic warming profile.

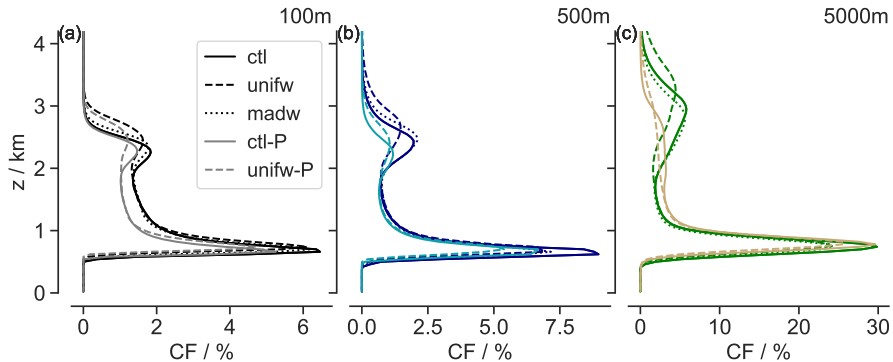

**Figure 10.** Profiles of second day domain averaged cloud fraction at three different horizontal resolutions (a-c) for all experiments.

With precipitation processes activated, the cloud field in a warmer climate responds with a cloud amount reduction across all resolutions, similar to that of the non-precipitating case, though the reductions in total cloud cover are slightly smaller 205 (Tab. 3 vs. Tab. 2). We are aware of two main mechanism that could be contributing to the dampening. First, precipitation has a constraining effect on cloud deepening, noted by Blossey et al. (2013) and Bretherton et al. (2013). At 500 m resolution the boundary layer deepening with warming is half and at 5 km only a third as much as in the non-precipitating simulations. Therefore, especially near the inversion changes in cloud fraction are reduced (Fig. 10). Second, evaporation of precipitation in the lower cloud layer counteracts drying. Vogel et al. (2016) who integrated for a longer time period reported likewise that 210 precipitation reduces deepening and drying with warming. In this way, precipitation is thought to promote the robustness of shallow cumulus clouds to warming. Regardless, though, we find the same dependency on resolution of how shallow cumulus cloud coverage responds to warming as in the non-precipitating simulations.

To summarize, the different experiments all exhibit the same horizontal resolution dependency on the representation and 215 response of shallow cumulus clouds to warming (Fig. 9). The resolution induced differences are larger than those between the different experimental setups. This confirms that horizontal resolution affects the representation and therewith response of shallow cumulus clouds to warming to first oder: the simulated shortwave cloud radiative feedback differs between the resolutions mainly in proportion to the basic state cloud fraction (Fig. 8) and therefore the cloud feedback strength increases at coarse resolutions. Hohenegger et al. (2020) who investigated grid spacings ranging from 2.5 km to 80 km found that cloud





cover increases up to 80 km horizontal resolution, which would, provided the results found here carry over also to even coarser resolutions, translate into further increased cloud feedback. At high resolutions, on the contrary, the trade wind cumulus cloud feedback converges to near-zero values.

## 5 Conclusions

This study explores the representation and response of shallow trade wind convection to warming and how that depends on
horizontal resolution by varying between 100 m and 5 km. Therewith we aim to bridge the gap between findings based on existing large eddy resolving simulations and emerging global storm resolving simulations. Based on the RICO case, simulations representative of trade wind conditions are compared to simulations with a 4 K warmed surface and atmosphere at constant humidity, representative of a simple idealized climate change. First, in a basic experiment the representation of shallow trade wind cumuli and their response to a uniformly warmed state is explored. Second, the sensitivity to resolution is probed in re-
fined experimental setups by including precipitation processes and warming along a moist adiabat in place of uniform warming.

At 100 m resolution a typical trade wind cumuli field is simulated that is in line with observations (Nuijens et al., 2009), and the range of LES analyzed in the RICO intercomparison case (van Zanten et al., 2011). Total cloud cover accounts to 15% in the non-precipitating and 13% in the precipitating case with a prominent peak in all cases near cloud base. At coarser res-
olutions, cloud cover is substantially enhanced and clouds are deeper; in the most extreme case at 5 km resolution total cloud cover is around 3.5 times more extensive. Cloud cover increases mostly due to enhanced cloudiness at cloud base. Weaker subcloud layer ventilation could explain the enhanced cloudiness and a weaker lateral entrainment rate allows the clouds to reach higher. Nevertheless, the overall structure of the boundary and cloud layer bear surprising similarity across resolutions explored here, suggesting that, although distorted, the same set of processes act in all cases.

In response to warming a cloud reduction can be observed consistently across resolutions. However, whereas at 100 m grid spacing the cloud reduction is rather small, at coarse resolutions the reductions are substantially enhanced. A robust dependency between cloud cover amount and its change with warming emerges: the more clouds are present in the control climate, the more cloud cover reduces in a warmer climate. Including precipitation processes mainly acts to limit the cloud layer deep-
ening by causing a net warming of the upper cloud layer and thereby stabilising the lower troposphere. A similar effect is found when the warming is done along a moist adiabat. These more refined setups result in nearly constant cloud top height with warming, questioning the idea that a cloud deepening is critical to a positive cloud cover feedback (Rieck et al., 2012). Regardless, the resolution dependence pertaining to cloud cover change is practically the same. On the contrary, a negative cloud optical depth feedback arises in all simulations due to an increasing cloud liquid water path. Although the magnitude of
this feedback varies, there is no obvious dependence on resolution. This is to be expected since increasing amounts of water vapor available for condensation with warming at constant relative humidity is a fundamental physical fact.

All in all, the compensation between the decreasing cloud cover and increasing cloud water with warming results in our case with convergence towards near-zero trade wind cumulus cloud feedback. Both of these effects appear physically appeal-

ing: a stronger vertical gradient in specific humidity results in a lowered relative humidity when mixing is activated, and all other things being equal in a slight reduction of the areal fraction where condensation can occur, whereas more availability of water vapor in the boundary layer results in thicker clouds. Provided the identified resolution-dependence of the cloud cover feedback carries over to other model codes, then it implies that storm resolving models may exaggerate trade wind cumulus cloud feedback. It is also interesting to compare with earlier studies, where LES simulations previously have suggested trade

wind cumulus feedback in the range 0.3 and 2.3 $\mathrm{Wm^{-2}K^{-1}}$ (Bretherton, 2015; Nuijens and Siebesma, 2019), and observational studies up until recently likewise 0.3 - 1.7 $\mathrm{Wm^{-2}K^{-1}}$ (Klein et al., 2017). A recent observational study, however, finds a near-zero trade wind cumulus cloud feedback (Myers et al., submitted), which is in line with our results.

*Code availability.* The ICON model source code is available for scientific use under an institutional or a personal non-commercial research license. Specific information on how to obtain the model code can be found under: https://code.mpimet.mpg.de/projects/iconpublic/wiki/

How_to_obtain_the_model_code.

*Author contributions.* The original idea of this study was conceived by CH and TM, whereas all simulations and most analysis was conducted by JR. All authors contributed to the writing.

*Competing interests.* The authors declare that they have no conflict of interest.

*Acknowledgements.* This study was supported by the Max-Planck-Gesellschaft (MPG) and computational resources were made available

by Deutsches Klimarechenzentrum (DKRZ) through support from Bundesministerium für Bildung und Forschung (BMBF). TM received funding from the European Research Council (grant no. 770765), the H2020 European Research Council (grant no. CONSTRAIN 820829). This work was a contribution to the Cluster of Excellence 'CLICCS - Climate, Climatic Change, and Society', contribution to the Center for Earth System Research and Sustainability (CEN) of Universität Hamburg. This study benefitted from discussions and technical support from Guido Cioni and Tobias Becker.

**Appendix A: Initial profiles and large scale forcing**

In the RICO case, van Zanten et al. (2011) constructed initial profiles as piecewise linear fits of radiosonde measurements up to a height of 4 km. As modification to the case defined by van Zanten et al. (2011), radiation is computed interactively to be able to calculate shortwave cloud radiative effects, which requires a model top at about 20 km in ICON-LEM. Below 4 km height,





initial profiles as in van Zanten et al. (2011) are applied, above they are expanded accordingly, mostly with piecewise linear extrapolation, see Table A1 for details. The free tropospheric lapse rate $\frac{d\theta}{dz}$ is calculated with

$$\frac{d\theta}{dz} = \frac{Q_R}{w(z)}, \tag{A1}$$

where the imposed subsidence $w$ balances a radiative cooling $Q_R$ of 2.5 Kday$^{-1}$ as suggested in the RICO setup. The temperature profile thus follows roughly a moist adiabat in the lower free troposphere. At 17 km, a tropopause of 195 K is included. The specific humidity profile is calculated from relative humidity following a linear decrease from 20% at 4 km height to 1% at 15 km and 0% at 17 km height.

**Table A1.** Fixed points for piecewise linear profiles of $\theta$, $q_v$, $u$, $v$, the subsidence rate $W$ and the large scale forcing of heat $\partial_t\theta|_{LS}$ and moisture $\partial_t q_v|_{LS}$ extended from the RICO case (van Zanten et al., 2011), from 4 km to 22 km height.

| Height | $\theta$ | $q_v$ | $u$ | $v$ | $W$ | $\partial_t q_v|_{LS}$ | $\partial_t\Theta|_{LS}$ |
| m | K | kgkg$^{-1}$ | ms$^{-1}$ | ms$^{-1}$ | ms$^{-1}$ | gkg$^{-1}$day$^{-1}$ | Kday$^{-1}$ |
|---|---|---|---|---|---|---|---|
| 0 | 297.9 | 0.016 | -3.8 | -9.9 | 0 | -1.0 | -2.5 |
| 740 | 297.9 | 0.0138 | | | | | |
| 2260 | 306.8 | | | | -0.005 | | |
| 2980 | | | | | | 0.3456 | |
| 3260 | | 0.0024 | | | | | |
| 4000 | | 0.0018 | | -1.9 | -0.005 | 0.3456 | |
| 5000 | | | | | -0.007 | | |
| 7000 | | | | | | 0.13824 | |
| 10000 | (A1) | q(rh) | | | | 0.03456 | |
| 12000 | | | | 16.1 | -0.007 | | -2.5 |
| 15000 | | q(rh=1) | | | | 0 | |
| 17000 | 381.03 | 0 | | | 0 | 0 | -0.4 |
| 22000 | | 0 | -3.8 | -1.9 | 0 | | 0 |

## Appendix B:  Impact of domain size

In order to confidently link the observed differences to characteristics of the resolution and not of the domain size, a simulation at the same horizontal resolution (1 km) is performed on two different domain sizes (50 km and 500 km). The simulations show that differences between the cloud field on the two domains are small (Fig. B1). With larger domain size, clouds are slightly deeper and show a narrower cloud fraction profile; total cloud cover is 1% points less (1 km resolution). On the same domain, the cloud cover would hence be even larger with the coarser resolutions.

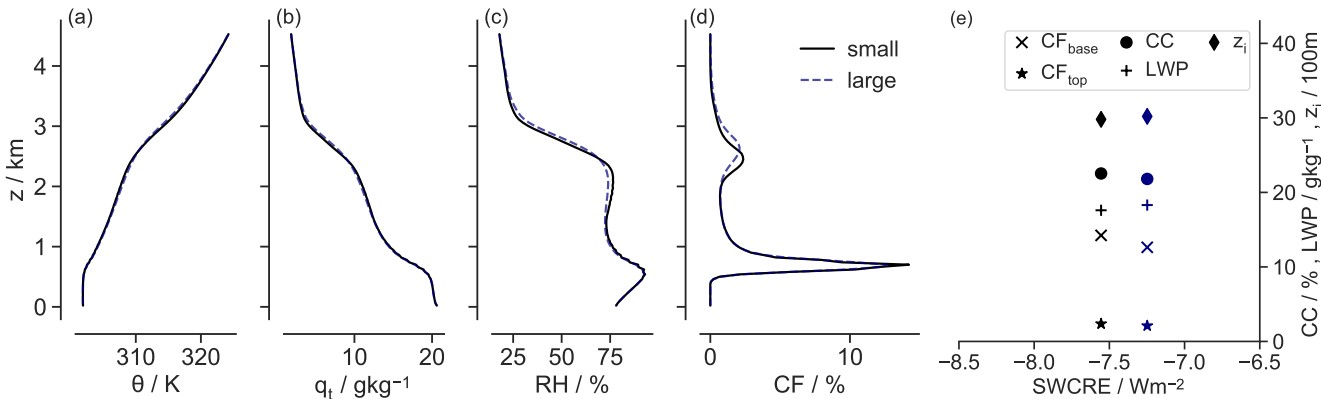

**Figure B1.** Profiles of second day domain averaged (a) potential temperature $\theta$, (b) specific humidity $q_t$ , (c) relative humidity $RH$ and (d) cloud fraction $CF$; as well as (e) mean values of total cloud cover ($CC$), cloud fraction at base ($CF_{base}$) and top ($CF_{top}$), liquid water path ($LWP$) and inversion height ($z_i$) at 1 km resolution on two different domain sizes: 50 x 50 km (black, small) and 500 x 500 km (blue-dashed, large).

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
