# Peer review of "Shallow Cumulus Cloud Feedback in Large Eddy Simulations - Bridging the Gap to Storm Resolving Models"

_Atmospheric Chemistry and Physics, 2020_

## Referee Comment (RC1) · Anonymous Referee #1 · 7 Dec 2020

The authors present a nice, clear and well worked out analysis looking to bridge the LES and convection permitting model scales in regards to subtropical cloud feedbacks. I think it is possibly worth adding to the conclusion 'SRMs may exaggerate the trade wind cumulus cloud feedback' that this for a SRM that is configured with a similar all-or nothing cloud scheme. It is true that most SRMs have an all or nothing scheme, but it would be interesting to get some commentary from the authors on this feature as SRMs become more the norm and include more complex cloud schemes. It would also be good to get a discussion from the others as to why their results point towards a zero cloud feedback in the trade cumulus, while previous studies do not.

[Figure]

Ln 29: the parenthetical statement ('unlike climate models') needs to be set off from the rest of the sentence.

Ln 40: Why allegedly?

Ln 76: I think maybe 'idealized'- while sort of technically correct this tends to be in relation to a person's beliefs (https://www.merriam-webster.com/dictionary/idealistic)

Ln 96: Does the trade inversion get set by the large scale forcing imposed on the LES? I see this is discussed later.

Ln 155: In the case of uniform warming I guess the inversion strength is held fixed- so we are only really looking at what is commonly characterized as the SST dependence of subtropical cloud (Klein et al., 2017). I see this is discussed on ln 195, but it might be useful to note this here.

Fig. 9: why aren't the unifw-P and madw simulations joined with a line?

Ln: 240: Can you comment on how much inter-LES differences in resolution might affect their inferred cloud feedback strength, or is there only a sizable shift when convection permitting simulations are examined? Also- on line 260 it is noted that other LES studies have inferred a substantial positive feedback. Can you comment on why these studies ended up with this result and the present study does not?

Klein, S. A., Hall, A., Norris, J. R., and Pincus, R.: Low-Cloud Feedbacks from Cloud-Controlling Factors: A Review, Surv Geophys, 38, 1307-1329, 10.1007/s10712-017-9433-3, 2017.
* * *

---

## Referee Comment (RC2) · Anonymous Referee #2 · 16 Dec 2020

Review of "Shallow Cumulus Cloud Feedback in Large Eddy Simulations — Bridging the Gap to Storm Resolving Models" by Radtke, Mauritsen and Hohenegger, ACP-2020-1160.

Summary:

An idealized shallow cumulus case based on the RICO field campaign is simulated at a number of horizontal resolutions with and without idealized warming perturbations. These simulation are performed in a single model, the ICON-LEM. The authors find that cloud fraction in the present day climate are proportional to grid spacing. Similarly, the decrease in cloud fraction with idealized warming perturbations is also proportional

to grid spacing. The highest resolution simulations suggest (very) small positive cloud feedbacks in response to warming. This result is robust to the inclusion of precipitation and the type of warming perturbation, i.e., uniform in the vertical or moist adiabatic.

==========================

Assessment:

The paper is focused and clearly written and illustrated with figures. I have only minor suggestions, mostly related to additional references and clarifications.

Recommendation: Minor revisions.

==========================

Minor comments/additional references:

1. Blossey et al (2009, JAMES, https://doi.org/10.3894/JAMES.2009.1.8) also includes a study of the effect of changes in horizontal and vertical grid spacing on shallow cumulus clouds in two dimensional simulation across a similar range of resolutions as in the present paper. See in particular section 4 and figure 8 of that paper. Those simulations were based on composite low cloud regimes from a superparameterized global simulation and sought to understand the robustness of a negative low cloud feedback in that model. While the sign of the cloud response to warming was different in that setting, the same message emerged as in the present paper: higher resolution led to smaller cloud fractions and a weaker cloud response to warming in one of the cases considered in that paper. The setup in those simulations was more complicated than here, including an adaptive large-scale vertical velocity based on the weak temperature gradient approximation.

2. I didn't understand from the paper whether the RRTM radiation computation was offline, meaning that the fluxes were computed but the heating rates not applied to the ICON LEM fields, or online, meaning that they were. The description on p3/l68-69 suggests that they were offline, because the Van Zanten et al large-scale forcings (which in

part represent radiative heating) are applied in the boundary layer. I would ask the authors to make this distinction clear in the text and also to refer the reader to the appendix for more details. Whether the radiation computation is online or offline, the results of the paper are worth publishing. If longwave radiative heating is not included in the simulations, I would ask the authors to mention Narenpitak and Bretherton (2019, JAMES, https://doi.org/10.1029/2018MS001572) who found a negative shallow cumulus feedback that was driven in part by stronger LW cooling of the trade cumulus BL in a warmer climate. See also Wyant et al (2009, JAMES, https://doi.org/10.3894/JAMES.2009.1.7) who earlier hypothesized this mechanism. Narenpitak and Bretherton also considered two resolutions, 100m and 4km, in their simulations.

============================

Specific comments (3/58 means p. 3, line 58):

3/58: It would be useful to the reader to know what cloud droplet number and/or aerosol concentration was specified for the precipitating simulations. This could be moved to the appendix if necessary.

5/106: Would it be worth citing Cheng et al (2010, JAMES, https://doi.org/10.3894/JAMES.2010.2.3) who looked at the effect of differing horizontal resolution in the present day setting? If there are multiple studies along these lines, it could be phrased as "(e.g., Cheng et al, 2010)".

7/123-130: Is there a good reference that talks about the changes in mixing in shallow clouds as resolution decreases?

8/139: It might be worth citing Albrecht (1993, JGR, https://doi.org/10.1029/93JD00027) just after "... because it removes moisture available for evaporation near the inversion."

10/180-188: I would suggest using the phrases "cloud amount feedback" and "cloud optical depth feedback" here, as is done later in the paper.

[Figure]

11/194: The uniform warming perturbation implies a negative change in EIS from present day to warmed climate. Following the climatological fit of Wood and Bretherton (2006), this would imply a decrease in cloud fraction is expected (if one assumes a similar relationship in a future climate). However, the response of this particular case does not seem to follow that prediction. This is just a comment and doesn't need to be acknowledged in the paper.

14/259-262: Presumably the shallow cumulus feedback will be an aggregate feedback over a number of cloud regimes, weighted by the frequency of occurrence of those cloud regimes (which itself could change with climate). The case presented in this paper predicts the cloud response in one of those cloud regimes. If the high-resolution cloud fraction and/or SWCRE differ from the observed mean in shallow cumulus regions, would the near-zero cloud feedback predicted by the present study be expected to carry over to those other regimes?

15/282: It would be good to be explicit that \partial_t \Theta in the table is equivalent to Q_R here in the text. The caption to the table also uses a small \theta in \partial_t \theta, while the table header uses \Theta. It would be good to be consistent.

===========================

Typographical suggestions:

7/136-137: suggested rephrasing: "... are very similar, the inversion height in the precipitating case with 500m and 5km resolution is around 150m and 350m lower, respectively, than in the non-precipitating case ...". I felt like this wording would be easier for the reader to follow.

12/212: Suggested re-wording: "... responds to warming in both precipitating and non-precipitating simulations." I think that the emphasis on "both" is helpful for the reader.

14/253-254: Suggested re-wording: "All in all, the decrease of cloud cover and increase in cloud water with warming compensate and result in convergence to a nearzero trade wind cloud feedback at high resolution in these simulations."

---

## Author Comment (AC1) · 7 Jan 2021

**Reply to reviewers: Shallow Cumulus Cloud Feedback in Large Eddy Simulations - Bridging the Gap to Storm Resolving Models**

Jule Radtke, Thorsten Mauritsen, Cathy Hohenegger

We thank the referees for their constructive comments, that have helped to improve the manuscript. In the following, referees' comments are in *italics*, authors' responses are in normal font.

**Anonymous Referee 1**

*The authors present a nice, clear and well worked out analysis looking to bridge the LES and convection permitting model scales in regards to subtropical cloud feedbacks.*

We thank the referee for the positive assessment.

*I think it is possibly worth adding to the conclusion 'SRMs may exaggerate the tradewind cumulus cloud feedback' that this for a SRM that is configured with a similar all-or nothing cloud scheme. It is true that most SRMs have an all or nothing scheme, but it would be interesting to get some commentary from the authors on this feature as SRMs become more the norm and include more complex cloud schemes.*

We agree this is worth adding. We rewrote line 257 "Provided the identified resolution-dependence of the cloud amount feedback carries over to other model codes, then it implies that storm resolving models configured with a similar all-or-nothing cloud scheme may exaggerate the trade wind cumulus cloud feedback."

*It would also be good to get a discussion from the others as to why their results point towards a zero cloud feedback in the trade cumulus, while previous studies do not.*

Please note our comment to this point below.

*Ln 29: the parenthetical statement ('unlike climate models') needs to be set off from the rest of the sentence.*

We added that.

*Ln 40: Why allegedly?*

True, we deleted that.

*Ln 76: I think maybe 'idealized'- while sort of technically correct this tends to be in relation to a person's beliefs (https://www.merriam-webster.com/dictionary/idealistic)*

Thank you for pointing this out, we modified that.

*Ln 96: Does the trade inversion get set by the large scale forcing imposed on the LES? I see this is discussed later.*

Yes, to clarify this we added in line 96: "... the trade wind inversion .... which develops as a result of the prescribed large scale subsidence."

*Ln 155: In the case of uniform warming I guess the inversion strength is held fixed- so we are only really looking at what is commonly characterized as the SST dependence of subtropical cloud (Klein et al., 2017). I see this is discussed on ln 195, but it might be useful to note this here.*

*Klein, S. A., Hall, A., Norris, J. R., and Pincus, R.: Low-Cloud Feedbacks from Cloud-Controlling Factors: A Review, Surv Geophys, 38, 1307-1329, 10.1007/s10712-017-9433-3, 2017.*

Yes, we added noting this here already with "... a uniform temperature shift, which implies a fixed inversion strength and is commonly characterized as the SST dependence (Klein et al. 2017), ...".

*Fig. 9: why aren't the unifw-P and madw simulations joined with a line?*

Too many lines make the plot quite busy. We think not showing the lines eases the understanding of the plot and makes no difference to its message. Therefore, we hesitate to change this.

*Ln: 240: Can you comment on how much inter-LES differences in resolution might affect their inferred cloud feedback strength, or is there only a sizable shift when convection permitting simulations are examined? Also- on line 260 it is noted that other LES studies have inferred a substantial positive feedback. Can you comment on why these studies ended up with this result and the present study does not?*

This is an inherently difficult question to answer. We added the following speculative text that rounds off the paper: "It is perhaps tempting to think that other LES studies were under-resolved, that is, if they had been run with higher resolutions their estimated cloud feedback might have decreased. Although it seems likely that most LES will exhibit a similar resolution dependence of the cloud amount feedback to that found here, it is not clear why they should all converge to a near-zero total feedback given their differences in e.g. microphysics, and so no conclusions in this regard can be drawn here. It is, however, an interesting question for the community to address in the future."

**Anonymous Referee 2**

*Review of "Shallow Cumulus Cloud Feedback in Large Eddy Simulations — Bridging the Gap to Storm Resolving Models" by Radtke, Mauritsen and Hohenegger, ACP-2020-1160.*

*Summary:*

*An idealized shallow cumulus case based on the RICO field campaign is simulated at a number of horizontal resolutions with and without idealized warming perturbations. These simulation are performed in a single model, the ICON-LEM. The authors find that cloud fraction in the present day climate are proportional to grid spacing. Similarly, the decrease in cloud fraction with idealized warming perturbations is also proportional to grid spacing. The highest resolution simulations suggest (very) small positive cloud feedbacks in response to warming. This result is robust to the inclusion of precipitation and the type of warming perturbation, i.e., uniform in the vertical or moist adiabatic.*

*Assessment:*

*The paper is focused and clearly written and illustrated with figures. I have only minor suggestions, mostly related to additional references and clarifications.*

*Recommendation: Minor revisions.*

We thank the referee for the positive assessment.

*Minor comments/additional references:*

*1. Blossey et al (2009, JAMES, https://doi.org/10.3894/JAMES.2009.1.8) also includes a study of the effect of changes in horizontal and vertical grid spacing on shallow cumulus clouds in two dimensional simulation across a similar range of resolutions as in the present paper. See in particular section 4 and figure 8 of that paper. Those simulations were based on composite low cloud regimes from a superparameterized global simulation and sought to understand the robustness of a negative low cloud feedback in that model. While the sign of the cloud response to warming was different in that setting, the same message emerged as in the present paper: higher resolution led to smaller cloud fractions and a weaker cloud response to warming in one of the cases considered in that paper. The setup in those simulations was more complicated than here, including an adaptive large-scale vertical velocity based on the weak temperature gradient approximation.*

Thank you for drawing our attention to that reference, we added in line 259: "Blossey et al. (2009), who also included a study of the effect of grid spacing on shallow cumulus clouds in two dimensional simulations, came to the same conclusion, while the setup was more complicated and the sign of the cloud response to warming was different: at higher resolution cloud fractions are smaller and the cloud response to warming weaker."

*2. I didn't understand from the paper whether the RRTM radiation computation was offline, meaning that the fluxes were computed but the heating rates not applied to the ICON LEM fields, or online, meaning that they were. The description on p3/l68-69 suggests that they were offline, because the Van Zanten et al large-scale forcings (which in part represent radiative heating) are applied in the boundary layer. I would ask the authors to make this distinction clear in the text and also to refer the reader to the appendix for more details. Whether the*

*radiation computation is online or offline, the results of the paper are worth publishing. If longwave radiative heating is not included in the simulations, I would ask the authors to mention Narenpitak and Bretherton (2019, JAMES,https://doi.org/10.1029/2018MS001572) who found a negative shallow cumulus feedback that was driven in part by stronger LW cooling of the trade cumulus BL in a warmer climate. See also Wyant et al (2009, JAMES, https://doi.org/10.3894/JAMES.2009.1.7) who earlier hypothesized this mechanism. Narenpitak and Bretherton also considered two resolutions, 100m and 4km, in their simulations.*

Thank you for recommending the references. Our RRTM radiation was computed online, we clarify this in line 67-70: "As modification to the case defined by van Zanten et al. (2011), radiation is computed interactively online to be able to calculate cloud radiative effects ... Below 4 km height, initial profiles and large scale forcings as in van Zanten et al. (2011), besides the radiative cooling, are applied, above they are expanded accordingly, mostly with piecewise linear extrapolation, see Appendix A1 for details."

*Specific comments (3/58 means p. 3, line 58):3/58:*

*It would be useful to the reader to know what cloud droplet number and/or aerosol concentration was specified for the precipitating simulations. This could be moved to the appendix if necessary.*

We corrected and clarified in line 58: "In experiments with precipitation processes a one-moment microphysics scheme including cloud water, rain, snow and ice with a constant cloud droplet concentration of $200 \, \text{cm}^{-3}$ (Doms et al., 2011) is applied."

*5/106 Would it be worth citing Cheng et al (2010, JAMES, https://doi.org/10.3894/JAMES.2010.2.3) who looked at the effect of differing horizontal resolution in the present day setting? If there are multiple studies along these lines, it could be phrased as "(e.g., Cheng et al, 2010)".*

Thank you for recommending this reference. We rewrote line 106/107: "Most importantly, we note that at coarser resolutions cloud cover is substantially enhanced (Fig. 2). This was similarly found in e.g. Cheng et al (2010)."

*7/123-130: Is there a good reference that talks about the changes in mixing in shallow clouds as resolution decreases?*

We are not aware there is a good reference.

*8/139: It might be worth citing Albrecht (1993,JGR,https://doi.org/10.1029/93JD00027) just after "...because it removes moisture available for evaporation near the inversion.*

We added that.

*"10/180-188: I would suggest using the phrases "cloud amount feedback" and "cloud optical depth feedback" here, as is done later in the paper.*

We followed this suggestion and rewrote lines 180-188: "This is due to a compensating cloud optical depth feedback ... In contrast to the cloud amount feedback, though, the cloud optical depth feedback is not strongly resolution dependent."

*11/194: The uniform warming perturbation implies a negative change in EIS from present day to warmed climate. Following the climatological fit of Wood and Bretherton(2006), this would imply a decrease in cloud fraction is expected (if one assumes asimilar relationship in a future climate). However, the response of this particular case does not seem to follow that prediction. This is just a comment and doesn't need to be acknowledged in the paper.*

Thank you for this comment.

*14/259-262: Presumably the shallow cumulus feedback will be an aggregate feedback over a number of cloud regimes, weighted by the frequency of occurrence of those cloud regimes (which itself could change with climate). The case presented in this paper predicts the cloud response in one of those cloud regimes. If the high-resolution cloud fraction and/or SWCRE differ from the observed mean in shallow cumulus regions, would the near-zero cloud feedback predicted by the present study be expected to carry over to those other regimes?*

There is ample evidence that stratocumulus and trade wind cumulus clouds behave differently to warming, and so we do not think stratocumulus clouds will exhibit a near-zero cloud feedback given sufficient resolution. Other regimes such as mid- and high latitude shallow cumulus or fair weather cumulus over land may be more similar to the regime studied here, but we prefer to not speculate on this point.

*15/282: It would be good to be explicit that $\partial_t \Theta$ in the table is equivalent to $Q_R$ here in the text. The caption to the table also uses a small $\theta$ in $\partial_t \theta$, while the table header uses $\Theta$. It would be good to be consistent.*

Thank you for pointing out this inconsistency, we adapted this.

*Typographical suggestions:*

*7/136-137: suggested rephrasing: "... are very similar, the inversion height in the precipitating case with 500m and 5km resolution is around 150m and 350m lower, respectively, than in the non-precipitating case ...". I felt like this wording would be easier for the reader to follow.*

Thank you for this suggestion. We followed the suggested rephrasing.

*12/212: Suggested re-wording: "... responds to warming in both precipitating and non-precipitating simulations." I think that the emphasis on "both" is helpful for the reader.*

Thank you for pointing this out. We followed the suggested re-wording.

[revised manuscript text omitted]